# Characteristics and Electronic Band Alignment of a Transparent *p*-CuI/*n*-SiZnSnO Heterojunction Diode with a High Rectification Ratio

**DOI:** 10.3390/nano11051237

**Published:** 2021-05-07

**Authors:** Jeong Hyuk Lee, Byeong Hyeon Lee, Jeonghun Kang, Mangesh Diware, Kiseok Jeon, Chaehwan Jeong, Sang Yeol Lee, Kee Hoon Kim

**Affiliations:** 1Center for Novel States of Complex Materials Research, Department of Physics and Astronomy, Seoul National University, Seoul 08826, Korea; xaaer5@snu.ac.kr (J.H.L.); kangjh1127@snu.ac.kr (J.K.); mangeshonnet@gmail.com (M.D.); 2Department of Microdevice Engineering, Korea University, Seoul 02841, Korea; endl0824@naver.com; 3Department of Chemical and Biomolecular Engineering, Yonsei University, Seoul 03722, Korea; jks8287@kitech.re.kr; 4Smart Energy and Nano Photonics R&D Group, Korea Institute of Industrial Technology, Gwangju 61012, Korea; chjeong@kitech.re.kr; 5Department of Electronic Engineering, Gachon University, Seongnam-si 13120, Korea; 6Institute of Applied Physics, Department of Physics and Astronomy, Seoul National University, Seoul 08826, Korea

**Keywords:** copper iodide (CuI), SiZnSnO (SZTO), transparent diode, current rectification ratio, energy band alignment

## Abstract

Transparent *p*-CuI/*n*-SiZnSnO (SZTO) heterojunction diodes are successfully fabricated by thermal evaporation of a (111) oriented *p*-CuI polycrystalline film on top of an amorphous *n*-SZTO film grown by the RF magnetron sputtering method. A nitrogen annealing process reduces ionized impurity scattering dominantly incurred by Cu vacancy and structural defects at the grain boundaries in the CuI film to result in improved diode performance; the current rectification ratio estimated at ±2 V is enhanced from ≈10^6^ to ≈10^7^. Various diode parameters, including ideality factor, reverse saturation current, offset current, series resistance, and parallel resistance, are estimated based on the Shockley diode equation. An energy band diagram exhibiting the type-II band alignment is proposed to explain the diode characteristics. The present *p*-CuI/*n*-SZTO diode can be a promising building block for constructing useful optoelectronic components such as a light-emitting diode and a UV photodetector.

## 1. Introduction

The γ-phase of copper iodide (CuI) with a direct bandgap (*E*_g_) of ≈3.1 eV has been attracting increased attention as an emergent transparent *p*-type semiconductor with a high hole mobility (*μ*_h_); for example, the *μ*_h_ value reaches up to ≈44 cm^2^V^−1^s^−1^ in a single crystal, thereby being higher than that of any other conventional *p*-type oxides [1]. Even in a thin film form, CuI exhibits a relatively high *μ*_h_ of typically 6–7 cm^2^V^−1^s^−1^ at a hole concentration (*p*) of ≈10^19^ cm^−3^ [2]. Moreover, with the additional heat treatment such as ex situ thermal annealing [3] or in situ thermal annealing during deposition [4], *μ*_h_ values of those films are further enhanced more up to ≈26 cm^2^V^−1^s^−1^. Such thermal annealing processes are known to suppress not only Cu vacancy (*V*_Cu_), which is the main source of acceptor in the CuI system, but also structural defects such as pinholes or grain boundaries. In this respect, the CuI thin films with proper heat treatment can play an important role of being a transparent *p*-type layer in various optoelectronic devices such as solar cells [5], a light-emitting diode [6], a field-effect transistor [7], and a UV photodetector [8,9].

In order to realize a transparent *pn* diode based on the CuI, we have been investigating various heterojunction diodes made of *p*-CuI and *n*-type semiconductors [2,8,9,10,11]; in our previous work, we have reported a transparent diode made of *p*-CuI/*n*-BaSnO_3−__δ_ (BSO) films and its carrier transport behavior [12]. The realized *p*-CuI/*n*-BSO diode exhibited a high current rectification ratio (*I*_F_/*I*_R_) of 6.75 × 10^5^ at an external voltage bias of ±2 V. On the other hand, a high-quality BSO film often requires an epitaxial growth on a proper substrate such as SrTiO_3_ (001), and its ideal growth temperature is rather high (≈800 °C). Therefore, for the applications in e.g., flexible UV photodetectors, it is desirable to test a heterojunction diode made of an alternative *n*-type film that can be grown at a relatively low-temperature condition comparable to that of the CuI film and without the constraint of epitaxy.

Along this line of reasoning, Yamada et al. have recently reported that a heterojunction diode made of *p*-CuI film and the well-known *n*-type amorphous semiconductor InGaZnO (IGZO) film exhibits a high rectification ratio of ≈10^9^ [8]. Independent of this promising approach, we have been focusing on a new *n*-type amorphous semiconductor SiZnSnO (SZTO) film that exhibits a high optical gap *E*_g_ larger than 3.7 eV [13] and a field-effect mobility of ≈38 cm^2^V^−1^s^−1^ [14]. It has been found that the *n*-type SZTO film maintains enhanced electrical stability as compared to the IGZO film [13], which is possibly due to reduced oxygen vacancies as controlled by Si concentration via strong Si–O bonding [15]. In addition, the *n*-type SZTO film has another advantage of being composed of non-toxic and abundant elements of Si and Sn. Therefore, we decided to adopt the SZTO as a transparent *n*-type layer to realize a new heterojunction diode with the *p*-CuI film.

Another potential merit of the *p*-CuI/*n*-SZTO diode lies in the capability of tuning the electronic energy diagram of the *n*-type amorphous ZnSnO layer by the control of Si concentration. Such a capability might provide an opportunity for improving the efficiency of a potential optoelectronic device such as a light-emitting diode. As an example, Baek et al. [6] have shown that the performance of the light-emitting diode made of *p*-CuI/*n*-Mg*_x_*Zn_1−*x*_O quantum dot (*x* = 0, 2.7, and 6 at %) has improved when the energy level difference between *n*-Mg*_x_*Zn_1−*x*_O and Al electrode is adjusted via the control of *x*. The imbalance between the injected hole and electron carriers at the interface of a light-emitting diode can potentially lead to the decreased device performance. Thus, the control of electron and hole injection ratios via a proper tuning of energy band diagram can be an effective way to improve performance of the given light-emitting diode.

Yet another promising application direction of the *p*-CuI/*n*-SZTO diode is a self-powered UV photodetector [16]. Recently, the photo-response of various *pn* diode structures made of *p*-CuI films (or nanoparticles) and various *n*-type materials are widely being tested; those *n*-type materials include IGZO films [8], CsPbBr_3_ crystals [9], ZnO:Au films [17], and ZnO films [18], and β-Ga_2_O_3_ single crystals [19]. Once realized, those photodetectors made of the *pn* diode are expected to have self-powered characteristics and reliable responsivity under UV light illumination [8,9,17,18]. Thus, the diode made of *p*-CuI and *n*-SZTO can be another promising platform for realizing a flexible, transparent UV photodetector with a capability of tuning electronic band diagram in the SZTO layer. To realize such a device, quantitative understandings on the diode performance and energy band alignment at the interface should be prerequisite.

With the above motivations, we here report the realization of transparent heterojunction diodes made of novel transparent polycrystalline *p*-CuI film and amorphous *n*-SZTO film that has resulted in a high rectification ratio *I*_F_/*I*_R_ of ≈10^7^. To investigate the diode characteristics, various diode parameters, such as ideality factor, reverse saturation current, and series resistance, etc., are estimated from the diode curve fitting based on the Shockley diode equation. We propose an energy band diagram of the *p*-CuI/*n*-SZTO diode exhibiting type-II heterojunction and attribute the high rectification ratio to the improved transport properties of the CuI film, resulting from reduced ionized impurity scattering of Cu vacancies and enhanced diode interface via reduced structural defects. Our results show that the diode made of *p*-CuI/*n*-SZTO films exhibits an excellent electrical performance as a transparent *pn* diode with potential tunability of the energy band diagram, which can be useful for realizing flexible, cost-effective optoelectronic devices such as UV photodetectors and light-emitting diodes.

## 2. Experimental Section

In_2_O_3_:Sn (ITO, film thickness *t* = 50 nm) and SZTO (*t* = 27 nm) films were deposited on boro-aluminosilicate glass substrates (EAGLE XG slim glass, Corning) (*t* = 0.5 mm) by the DC and RF sputtering technique. The ITO ceramic disk was used as a target for the ITO film growth at a power of 30 W in Ar pressure of 4 mTorr at room temperature by a commercial DC magnetron sputtering (KVS-2002, Korea Vacuum Tech, Korea). After patterning with the photolithography methods, the SZTO film was subsequently deposited at a power of 60 W by the RF magnetron sputtering technique (KVS-2004, Korea Vacuum Tech, Korea) in a mixed gas of Ar and O_2_ at room temperature; the gas flow ratio of Ar and O_2_ was set for 40:1 sccm. The as-deposited SZTO/ITO film was annealed at 500 °C for 2 h at ambient condition, and the detailed conditions for the target synthesis and the film growth were described in our earlier report [13]. A custom-made thermal evaporator was used for the deposition of the CuI films with the CuI powder (purity ≈99.998%) at room temperature on glass substrates (soda-lime glass for a microscope slide, Marienfeld) and SZTO/ITO/glass. For the fabrication of *p*-CuI/*n*-SZTO diodes, both CuI and Au/Ni films have been sequentially deposited with stencil masks that can produce 10 circular dots with a typical radius of 50 μm. The Au/Ni film (Figure 1a) was deposited to form ohmic contact between the CuI film and a tungsten probe tip; the Ni film (*t* = 5 nm) with a circular dot shape was first deposited on top of the CuI film, followed by the deposition of Au film (*t* = 50 nm) with the same circular dot size. It confirmed that two circular dots of the Au/Ni film grown on top of the CuI film exhibit linear behaviors in the current–voltage (*I*-*V*) curves by two tungsten probe tips connected to each dot (see, Appendix A). This implies that good ohmic contact was formed between Au/Ni films and CuI films as the *p*-type channel for a diode. Furthermore, the same measurements were performed with two tungsten probe tips connecting two respective circular dots of the ITO film grown on the SZTO film. It shows that good ohmic contact was formed between ITO and SZTO films as the *n*-type channel. The as-grown diodes were annealed at 50 °C for 125 h under the N_2_ gas flowing condition of 20 mL/min. Optical transmittance and reflectance spectra measurements on the CuI/SZTO/ITO/glass were carried out by the UV-VIS-NIR spectrophotometer (Cary 5E, Varian). The crystallinity of films has been investigated by the *θ*-2*θ* scans in a high-resolution X-ray diffractometer (Empyrean^TM^, PANalytical, Korea). The structural characterizations were conducted using atomic force microscopy (AFM) (NX10, Park Systems, Korea). The current density–voltage (|*j*|-*V*) (or *I*-*V*) characteristics of the diodes were investigated by the semiconductor parameter analyzer (4200-SCS, Keithley, Korea); all the measurements were performed under dark and air conditions at room temperature.

## 3. Results and Discussion

Figure 1a shows a schematic structure of the fabricated *p*-CuI/*n*-SZTO diode for electrical measurements, in which the SZTO film is grown on the ITO/glass, followed by the subsequent deposition of the circular CuI film and the metallic (Au/Ni) electrode. Figure 1b is an actual photograph of the CuI/SZTO/ITO/glass film without the Au/Ni layer, demonstrating that all the layers are transparent in the visible spectral range. The CuI/SZTO films (the orange dashed line) were deposited with a lateral size of 10 × 4 mm^2^ on the ITO/glass substrate (the green dashed line); the thicknesses of the CuI, the SZTO, and the ITO films are 110, 27, and 50 nm, respectively. Hereafter, we denote the term CuI (*x* nm) as *x* nm-thick CuI film where *x* represents the thickness of the CuI film (*t*_CuI_).

Figure 1c exhibits the optical transmittance (*T**_λ_*), reflectance (*R*), and absorption (*A*) spectra as a function of wavelength *λ* for the CuI/SZTO/ITO/glass (Figure 1b), and a black solid line represents *T**_λ_* of a bare glass substrate. The absorbance is obtained from the relationship of *A* = 100−*T**_λ_*−*R*, exhibiting small values less than 5% at *λ* ≥ 450 nm. The maximum *T**_λ_* of the CuI/SZTO/ITO/glass is indeed found to be as high as ≈90% and is always larger than ≈80% in the visible spectral region (500–750 nm). In addition, the transmittance of the glass substrate *T_λ_**_,_*_glass_ is a nearly constant of ≈0.9 (the black solid line). According to the relationship of *T_λ_*_,film_ = *T_λ_*_,measured_/*T_λ_*_,glass_, *T_λ_*_,film_ of the remaining films (CuI/SZTO/ITO) is then likely to be higher than ≈88% in the visible spectral region. All these results quantitatively demonstrate why the fabricated diode structure (CuI/SZTO/ITO/glass) maintains optical transparency, as shown in Figure 1b.

In addition, it is noteworthy that an exciton absorption peak is found at 407 nm (3.05 eV) (a gray dotted line in Figure 1c), as frequently observed in the CuI film [2,4,10,12]. Then, *E*_g_ of the CuI film is estimated as ≈3.1 eV from the relationship of *E*_g_ = *E*_Z1,2_ + *E*_ex_ where *E*_Z1,2_ is an exciton absorption energy of CuI and *E*_ex_ is an exciton binding energy of 62 meV [20]. The obtained *E*_g_ value is similar to our previous result (≈3.08 eV) [12] and other reports [2,3,4]. Based on the observation of the similar optical bandgap, the optical spectra are likely to be expected as high enough regardless of *t*_CuI_, as confirmed in the previous report [12]. Furthermore, the observation of such a sharp exciton absorption peak supports that the as-grown CuI film is of high quality.

The crystallinity of the as-grown films (CuI/SZTO/ITO/glass substrate) has been investigated by the X-ray *θ*-2*θ* scan in Figure 1d. The CuI film exhibits a preferred orientation along the (111) plane, similar to the CuI/glass sample case (see, Appendix A for the X-ray result). Since the nearest bottom layer is composed of an amorphous material, i.e., SZTO film or glass substrate, this (111) peak is unlikely to originate from the epitaxial CuI film [12]. It is well-known that the CuI system exhibits randomly oriented in-plane domains within the CuI (111) plane due to the low surface stability energy of the (111) plane [2]. The SZTO film cannot be identified in Figure 1d due to its amorphous nature, as confirmed by our previous transmission electron microscopy and X-ray photoelectron spectroscopy studies [21,22]. The extra peaks are the polycrystalline ITO film denoted as a symbol of *I* in Figure 1d.

To investigate the electrical transport of the diodes before and after the N_2_ annealing, we measured the *I*-*V* curves of the diodes, which have the structure shown in Figure 1a using a two-point contact method. In this configuration, the two probe tips are connected to Au/Ni and ITO films for *p*-CuI and *n*-SZTO layers, respectively. Figure 2a shows the *I*-*V* curve (black solid line) of the as-grown CuI (140 nm)/SZTO diode in a linear scale. A turn-on voltage (*V*_turn-on_), which is the on-set voltage for a large increase in the current level, has been determined from a linear extrapolation of the *I*-*V* curve in Figure 2a. The estimated *V*_turn-on_ of the as-grown diode was 1.73 V. Figure 2b presents the corresponding |*j*|-*V* curve in a semi-log scale. To extract the quantitative information on the diode parameters, we have tried to fit the data with the Shockley diode equation in the full range from −2 V to +2 V:(1)$$I\left(V\right)=I_{S}\left[\exp\left\{\frac{e\left(V-IR_{S}\right)}{\eta k_{B}T}\right\}-1\right]+\frac{V-IR_{S}}{R_{p}}+I_{0}$$
where *I*_s_ is the reverse saturation current, *η* is the ideality factor, *k*_B_ is the Boltzmann constant, *T* is the absolute temperature, *I*_o_ is the offset current, *R*_s_ is the series resistance, and *R*_p_ is the parallel resistance [11]. The orange lines in Figure 2b represent the curve fitting results, which indeed well explain the measured data except for the irreversible charge trap effect near 0.8 V. We will discuss the resultant parameters in Figure 3.

Similarly, the measured *I*-*V* and |*j*|-*V* curves in the annealed CuI (140 nm)/SZTO diode are plotted as red solid lines in Figure 2c,d, respectively. The green dashed line of the |*j*|-*V* curve represents the curve fitting result. What is most conspicuous in the behavior of the annealed diode is the decrease (increase) of the current level in the negative (positive) bias region. This immediately points out that the current rectification ratio *I*_F_/*I*_R_, where *I*_F_ is the current value at +2 V and *I*_R_ is the current at −2 V, is enhanced after the annealing. As the SZTO layer is expected to be stable at the annealing condition (50 °C, 125 h), to understand the enhancement of *I*_F_/*I*_R_, one should understand first how the annealing process will affect the properties of CuI film. It is known that in the as-grown CuI film, *V*_Cu_ is likely to exist, behaving as a dominant acceptor source and also as an ionized impurity scattering source. Upon the ex situ thermal annealing at 50 °C being performed for 125 h, iodine vacancies (*V*_I_) can be additionally created, thereby compensating the native *V*_Cu_. In addition to *V*_Cu_ compensation by *V*_I_, thermal energy can provide extra energy for rendering the migration of existing Cu vacancies from the inside of the CuI film to the outside of the surface or grain boundary [3]. Therefore, the optimal thermal annealing can lead to suppression of ionized impurity scattering via the compensation of *V*_Cu_ by *V*_I_ and also by the reduction of *V*_Cu_ inside the CuI film. At the same time, a reduction of structural defects near the grain boundaries of the CuI film is also expected, which in turn can form a smooth and uniform interface to improve the *pn* diode performance. Reduced grain boundaries after annealing can be verified in the AFM image of the CuI film/glass in Appendix A.

Several pieces of evidence supporting improved diode performance after annealing can be found in the *I*-*V* (or |*j*|-*V*) characteristics in Figure 2c,d; the hysteresis of the |*j*|-*V* curve between the forward and reverse sweeping directions decreases, *V*_turn-on_ slightly increases from 1.73 to 1.74 V, the offset current *I*_o_ decreases in the flat range from −2 to +0.7 V, and the |*j*|-*V* curve produces significant noise in the flat region. To better understand microscopically how the diode performance improves, we discuss here the supporting evidence in more detail. First, it should be reminded that irreversible charge traps mainly cause the diode hysteresis because the charged trap sites, such as *V*_Cu_, act as a slowly responding component under external voltage bias. Thus, the decreased hysteresis in the |*j*|-*V* curve demonstrates that the charged trap sites have been greatly reduced.

Second, the slight increase of *V*_turn__-on_ from 1.73 to 1.74 V also supports the improved diode performance. A threshold voltage (*V*_th_), which is an ideal voltage to begin the current flow, is expected to become fairly high in this diode; it is expressed by *V*_th-p_ = *V*_bi_ + |∆*E*_V_/*e*| for a *p*-type carrier and by *V*_th-n_ = *V*_bi_ + |∆*E*_C_/*e*| for an *n*-type carrier, where *V*_bi_ is the built-in potential of a diode, ∆*E*_V_ is the valence band offset, and ∆*E*_C_ is the conduction band offset between two semiconductors [2]. Based on the band diagram of the CuI/SZTO diode (vide infra, Figure 5), it turns out to be *V*_bi_ = 0.99 V, |∆*E*_V_/*e*| = 2.55 V, and |∆*E*_C_/*e*| = 1.90 V, resulting in *V*_th-p_ = 3.54 V for the injection of holes from *p*-CuI to *n*-SZTO and *V*_th-n_ = 2.89 V for the injection of electrons from *n*-SZTO to *p*-CuI. Thus, it is likely that electron injection is more probable than hole injection because the absolute conduction band offset |∆*E*_C_/*e*| is 0.66 V less than |∆*E*_V_/*e*|. The obtained *V*_turn-on_ = 1.74 V in Figure 2c for the annealed diode is obviously smaller than the predicted *V*_th-n_ = 2.89 V. Since the *V*_Cu_ creates an intermediate state that can form an additional current path at the diode interface, the actual *V*_turn-on_ should be lower than *V*_th_ in general. Therefore, in turn, the increased *V*_turn-on_ after annealing indicates reduced extra leakage current paths by the decrease of *V*_Cu_ or by the compensation of *V*_Cu_.

Finally, the reduced *I*_o_ after annealing also supports the improved diode performance. *I*_o_ can be generated by other leakage sources irrelevant to the bias voltage such as instrumental effects or pinholes that can exist across two electrodes. Once *I*_o_ was formed at a diode, it appears as a constant current in the flat region of the |*j*|-*V* curve; for instance, the *I*_o_ level is ≈5.9 pA in the bias range from −2 to +0.7 V in Figure 2b. Therefore, the reduced *I*_o_ level from 5.9 to 1.1 pA after the annealing in Figure 2d indicates that such a side effect has been mitigated by the annealing, too. As the *I*_o_ decreased after annealing, the current noise in the negative bias range concurrently increased. It implies that the actual current is small enough to reach the resolution limit of the instrument (Keithley 4200-SCS, ±1 pA).

Figure 3 compares all the obtained diode parameters for both CuI (20 nm)/SZTO and CuI (140 nm)/SZTO diodes before and after annealing. Here, we remind the physical meaning of each diode parameter except for *I*_F_/*I*_R_ and *I*_o_ already explained above. *I*_s_ represents the reverse saturation current, which occurs by the minority carriers located at the depletion region of a *pn* diode. *η* describes a diode transport model; *η* is known to change by diffusion (*η* = 1), recombination (*η* = 2), and numerous defects region (*η* > 2). *R*_s_ is determined by the contact resistance of several junctions between two films and between the electrode/film, and *R*_p_ represents the parasitic parallel resistance formed by an additional leakage path. In the case of the CuI (140 nm)/SZTO diode, the annealing process results in the increase of *I*_F_/*I*_R_ from 2.7 × 10^6^ to 6.6 × 10^7^ by ≈25 times, the decrease of *I*_o_ from 5.9 to 1.1 pA, the nearly same *I*_s_ being a quite low level of ≈0.07 fA, the decrease of *η* from 2.96 to 2.66, the decrease of *R*_s_ from 6.1 to 3.9 kΩ, and the slight increase of *R*_p_ from ≈2 to ≈5 TΩ shown in Figure 3.

The thermal annealing process, as explained in Figure 2, is likely to induce several physical processes; (1) the decrease of charged trap sites by *V*_Cu_, (2) a reduction of grain boundaries from the out-diffusion of *V*_Cu_, and (3) a reduced ionized impurity scattering with reduced *p*-type carriers (see, Appendix A for the CuI/glass Hall effect results). All the processes (1)–(3) can result in a decreasing tendency, as observed in the parameters *I*_o_ and *R*_s_. The decrease of *I*_o_ implies that other leakage sources such as the pinholes across two electrodes are reduced as a result of reduced vacancies, *p*-type carriers, and grain boundaries. Furthermore, with reduced grain boundaries, the contract resistance *R*_s_ between the CuI and the Au/Ni film is likely reduced, too. In addition, the *η* = 2.96 before annealing (Figure 3d) suggests the presence of numerous defects at the diode depletion region. Thus, the decrease of *η* from 2.96 to 2.66 after thermal annealing suggests reduced *V*_Cu_ and improved diode interface to become more homogeneous. The increase of *R*_p_ by ≈3 TΩ, albeit having a large least square fitting error close to the instrumental measurement limit, is also consistent with the reduced *V*_Cu_. The thermal annealing did not affect the saturation current, producing the nearly same value of *I*_s_ = ≈0.07 fA, which is close to the instrumental resolution limit. This implies that *I*_s_ = ≈0.07 fA is indeed close to an ideal value expected in this diode configuration. All the positive effects reflected in those parameters in Figure 3b–f after the annealing process coherently explains why *I*_F_/*I*_R_ has been improved by ≈25 times from 2.7 × 10^6^ to 6.6 × 10^7^ in the CuI (140 nm)/SZTO diode (Figure 3a).

To compare how the annealing can affect the characteristics of the diode made of a thinner CuI film, we have similarly studied the electrical properties of the CuI (20 nm)/SZTO diode before and after applying the same thermal annealing conditions, of which *I*-*V* (|*j*|-*V*) curves are provided in Appendix A. All the parameters obtained from the curve fitting results based on Equation (1) are also summarized in Figure 3; in the CuI (20 nm)/SZTO diode, the thermal annealing results in the decrease of *I*_o_ from ≈4.5 to 1.2 pA, the decrease of *I*_s_ from 20.5 fA to 0.20 fA, the decrease of *η* from 3.89 to 3.15, and the nearly same of *R*_p_ of ≈4 TΩ. The initial values of *I*_s_ and *η* after the growth seem to be all larger than the corresponding values in the CuI (140 nm)/SZTO diode, while *I*_o_ and *R*_p_ are comparable. This indicates that the contribution of *V*_Cu_ and related grain boundaries are larger in the thin (20 nm) films to cause the numerous defect regime in the diode, i.e., *η* = 3.89. After annealing, all the parameters of *I*_o_, *I*_s_, and *η* exhibit improvement, as similarly observed in the thick diode. Therefore, the thermal annealing seems to be also effective in improving the physical properties of the CuI (20 nm) film by reducing *V*_Cu_, grain boundaries, and ionized impurity scattering.

The most notable difference between the CuI (20 nm) and the CuI (140 nm) diode is found in the variation of *R*_s_ and *I*_F_/*I*_R_; after annealing, *R*_s_ increases from 95.8 to 133 kΩ, while *I*_F_/*I*_R_ exhibits a slight increase from 5.5 × 10^5^ to 2.4 × 10^6^. Note that *I*_F_(+2V)/*I*_R_(−2V) was calculated from the fitting curves in this case to avoid the noise-induced errors in the estimation of *I*_R_(−2V). In general, *R*_s_ is a main limiting factor for *I*_F_, as the forward bias voltage dropped by the interface contact resistance; *I*_F_ is supposed to decrease, being roughly proportional to *R*_s_. Therefore, the moderate increase of *I*_F_/*I*_R_ by a factor of ≈4 in the CuI (20 nm) diode, as compared with the factor of ≈25 increase in the CuI (140 nm) diode should be mainly attributed to the unexpected increase of *R*_s_. Without the deterioration of *R*_s_, the improvements of the other parameters *I*_o_, *I*_s_, and *η* should have produced much more enhanced *I*_F_/*I*_R_. The undesirable increase of *R*_s_ upon annealing indicates that the CuI (20 nm) film just starts to degrade due to the formation of a large amount of *V*_I_. The thermal degradation of the CuI has been known in the thin films with *t*_CuI_ < 100 nm [3] and even in a crystal [23]. If the CuI is subject to excessive thermal annealing, out-diffusion of *V*_Cu_ and the excessive formation of *V*_I_ can give rise to degradation in the physical properties of CuI; it becomes porous, exhibits yellowish-brown colors, and eventually develops many cracks [3]. Therefore, for a very thin film limit of *t*_CuI_ < 100 nm, it is expected that a shorter time or a lower temperature for annealing might be suitable to achieve a higher *I*_F_/*I*_R_. Thus, our results indicate that an optimal annealing condition should be carefully searched for each thin film with a different thickness.

It is emphasized that *I*_F_/*I*_R_ = 6.6 × 10^7^ is quite a high value among the heterojunction diodes consisting of the *p*-CuI film. Figure 4 compares *I*_F_/*I*_R_ of the *p*-CuI/*n*-SZTO diodes with those of other CuI-based transparent heterojunction diodes that have a high *I*_F_/*I*_R_ > ≈10^5^; 2 × 10^7^ at ±2 V for *p*-CuI/*n*-ZnO (a polycrystalline CuI film) [2], 2 × 10^9^ at ±2 V for *p*-CuI/*n*-ZnO (an epitaxial CuI film) [11], 7 × 10^5^ at ±2 V for *p*-CuI/*n*-BaSnO_3−__δ_ [12], 6 × 10^6^ at ±2 V for *p*-CuI/*n*-IGZO [3], and ≈10^9^ at ±1.5 V for *p*-CuI_1−*x*_Br*_x_*/*n*-IGZO (*x* = 0.0 − 1.0) [10]. Albeit the highest value of *I*_F_/*I*_R_ = ≈2 × 10^9^ realized in the *p*-CuI/*n*-ZnO diode is higher than that of the present *p*-CuI/*n*-SZTO diode by ≈30 times, to realize such a diode requires a special growth condition, i.e., an epitaxial growth of the CuI film on top of an epitaxial ZnO film. This implies that such a high *I*_F_/*I*_R_ is hard to achieve in a flexible device. Although the *I*_F_/*I*_R_ = ≈1 × 10^9^ in the *p*-CuI/*n*-IGZO diodes looks also higher than that of the *p*-CuI/*n*-SZTO diode, a proper comparison of *I*_F_/*I*_R_ might be required as the work used the measured *I*_F_ and the fitted *I*_s_ [10].

To quantitatively compare the performance of various diodes made of the *p*-type CuI and *n*-type materials, several diode parameters and photo-response results are also summarized in Table 1. The main parameter that characterizes the one-way electrical transport of a diode is *I*_F_/*I*_R_. Most of the diode exhibit *I*_F_/*I*_R_ larger than ≈7 × 10^5^ except recent two devices with rather large interface areas as they focus on the increase of photocurrent in the *pn* diode [9,19]. The ideality factor *η* seems to be lower than 2 when the *n*-type materials are epi-films of ZnO or amorphous IGZO film while the *η* of the *p*-CuI/*n*-SZTO is as large as 2.7–3.0, indicating that disorder effects in the CuI film can be further reduced by better annealing or growth conditions. The *I*_s_ of the *p*-CuI/*n*-SZTO diode is much lower than those of the *p*-CuI/*n*-ZnO and the *p*-CuI_1−*x*_Br*_x_*/*n*-IGZO diodes due to the inclusion of offset current in the fitting process. The *R*_s_ value of the *p*-CuI/*n*-SZTO is still higher than those of other diodes, leaving rooms for improvement by reducing interface resistance at the *pn* junction and *p*- or *n*-type films/electrodes. Therefore, if the *R*_s_ and the *η* can be further improved in the *p*-CuI/*n*-SZTO, the *I*_F_/*I*_R_ is likely to be enhanced more. Finally, in Table 1, to illustrate a direction of future applications, we have also listed several recent cases where the photo responsivity has been tested in the transparent *pn* diode made of the *p-*CuI. Based on these recent related studies, it is expected that flexible, optically transparent, cheap UV photodetectors can be fabricated by the *p*-CuI/*n*-SZTO diode.

Figure 5a compares the energy band alignment of several wide bandgap materials, including CuI [2,12], SZTO [13,22], IGZO [8,24], BaSnO_3_ [12,25], and ITO [26]. Based on the experimental results from previous reports, the band energies of those materials are aligned at the vacuum energy level *E*_vac_ = 0. The top (red) and bottom (blue) columns present conduction and valance bands, respectively. In order to investigate the energy band diagram of the annealed CuI (140 nm)/SZTO diode shown in Figure 5b, first of all, several energies such as *E*_vac_, valance band maximum (*E*_V_), and conduction band minimum (*E*_C_) are determined from Figure 5a. Subsequently, the Fermi energy level (*E*_F_) is calculated from the Anderson diode model for heterojunction [27]; various input parameters of valance band maximum *E*_V_, conduction band minimum *E*_C_ [2,13], dielectric constant [2,22], and effective masses [22,28] of both CuI and SZTO are adopted from the literature. *E*_g_ and *n* of SZTO are estimated from our previous work [13], while *E*_g_ and *p* of CuI are obtained in this work. Basic parameters and related references are summarized in Table 2.

The resultant band diagram is consistent with the type-II band alignment (or staggered gap type), in which *E*_C_ of the *n*-type material is located at the energy window between *E*_C_ and *E*_V_ of the *p*-type material. The predicted *eV*_bi_, the depletion width of CuI, and the depletion width of SZTO are found as 0.99 eV, 0.03 nm, and 37.1 nm, respectively. The present CuI/SZTO diode is indeed the one-side abrupt junction, of which the depletion region is mostly formed at, in this case, the SZTO layer. It is noted that the *n*-type depletion width of 37.1 nm is larger than the thickness of the SZTO film itself (27 nm). In this condition, an effective negative bias might be formed at the interface, and consequently, a finite forward bias should be required to reduce the electric field and to start the current flow across the *pn* diode. Consistent with this scenario, relatively high voltages for the current onset in the |*j*|-*V* curves have been indeed found at ≈0.7 V in Figure 2b,d.

## 4. Conclusions

We have studied the electrical characteristics of transparent *p*-CuI/*n*-SiZnSnO (SZTO) heterojunction diodes with a variation of CuI film thicknesses (20 nm and 140 nm), for which thermal evaporation and RF magnetron sputtering techniques have been utilized to deposit the (111) oriented CuI and amorphous SZTO films, respectively. Upon applying a thermal annealing condition of 125 h at 50 °C under N_2_ gas atmosphere, we have found that the current rectification ratio (*I*_F_/*I*_R_) is enhanced by ≈25 times up to 6.6 × 10^7^ in the thick CuI (140 nm)/SZTO diode due to the improvement of all the diode parameters as systematically obtained by the curve fitting to the Shockley diode equation. On the other hand, the thin CuI (20 nm)/SZTO diode exhibits a moderate increase of *I*_F_/*I*_R_ by ≈4 times up to 2.4 × 10^6^ due to the increase of contact resistance of the electrode. Changes of the diode parameters with a variation of the *p*-CuI film thickness or by the thermal annealing and their physical implications have been discussed in the two kinds of the transparent *p*-CuI/*n*-SZTO diode. Based on the comparison with the energy band alignment of various semiconductors, we have proposed the realization of the type-II band diagram in the *p*-CuI/*n*-SZTO diode. The present *p*-CuI/*n*-SZTO diode can be potentially used as a valuable transparent component in various optoelectronic applications such as a light-emitting diode or a UV photodetector.

## Figures and Tables

**Figure 1 nanomaterials-11-01237-f001:**
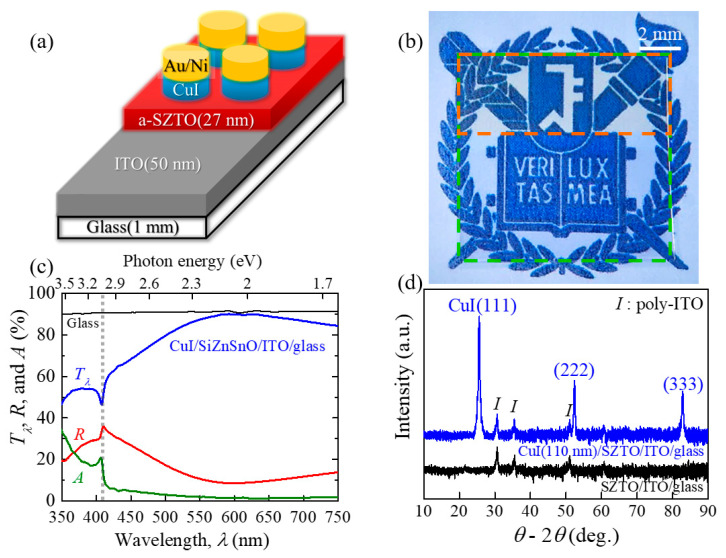
(**a**) A schematic structure of the *p*-CuI/*n*-SiZnSnO (SZTO) diode grown on the ITO deposited glass substrate, which was used for electrical characterization in this work. The CuI film of circular disk shape (with a diameter of 50 μm) were fabricated in two thicknesses of 140 and 20 nm while the thicknesses of SZTO and ITO films were 27 and 50 nm, respectively. Au/Ni (*t* = 50/5 nm) films were subsequently deposited on top of the CuI disk. (**b**) A separate film fabricated for optical characterization. The CuI(110 nm)/SZTO(27 nm) (the orange dashed line) were homogeneously deposited on top of the ITO/glass substrate (the green dashed line). All the regions exhibit clearly optical transparency. (**c**) Optical transmittance (*T_λ_*), reflectance (*R*), and absorption (*A*) spectra of the CuI (110 nm)/SZTO/ITO/glass sample shown in (**b**). The black solid line represents *T_λ_* of a glass substrate (*t* = 1 mm) only. (**d**) X-ray *θ*-2*θ* scan results of the CuI (110 nm)/SZTO/ITO/glass film shown in (**b**) and of another SZTO/ITO/glass film. The CuI film exhibits the preferential orientation of a (111) plane, and additional peaks represent a polycrystalline ITO film indicated as *I*.

**Figure 2 nanomaterials-11-01237-f002:**
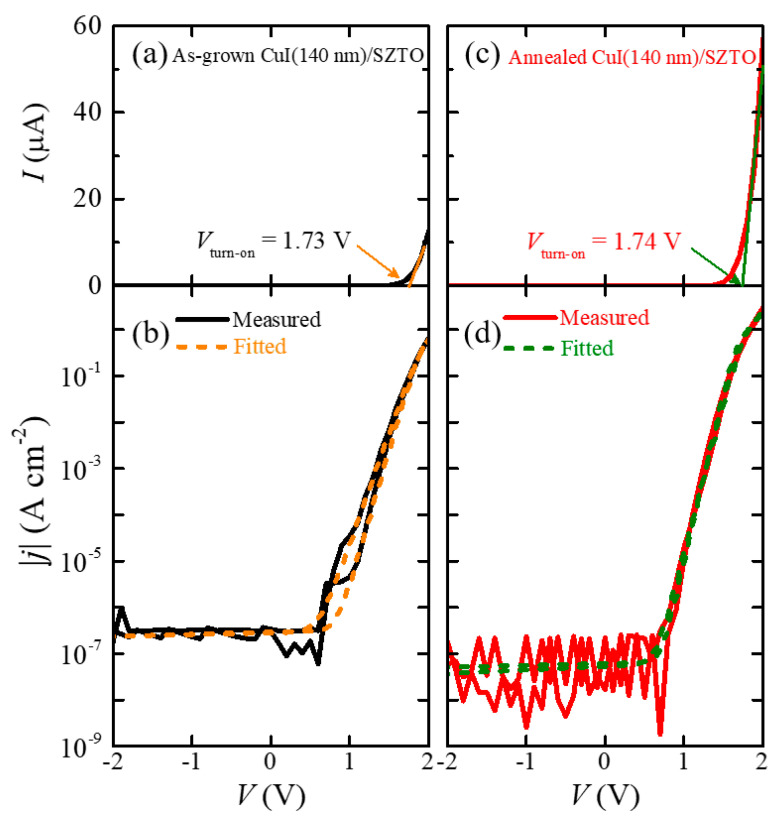
(**a**) The current–voltage *I*-*V* curve in a linear scale and (**b**) the current density–voltage *|j|-V* curve in a semi-log scale of the as-grown CuI (140 nm)/SZTO diode. (**c**) The *I*-*V* and (**d**) *|j|-V* curves of the annealed CuI(140 nm)/SZTO diode. The orange and the green solid lines in the *I*-*V* curves represent the linear extrapolation to determine the turn-on voltage of a diode, *V*_turn-on_, while the orange and green dashed lines in the *|j|-V* curves correspond to the diode fitting results based on Equation (1).

**Figure 3 nanomaterials-11-01237-f003:**
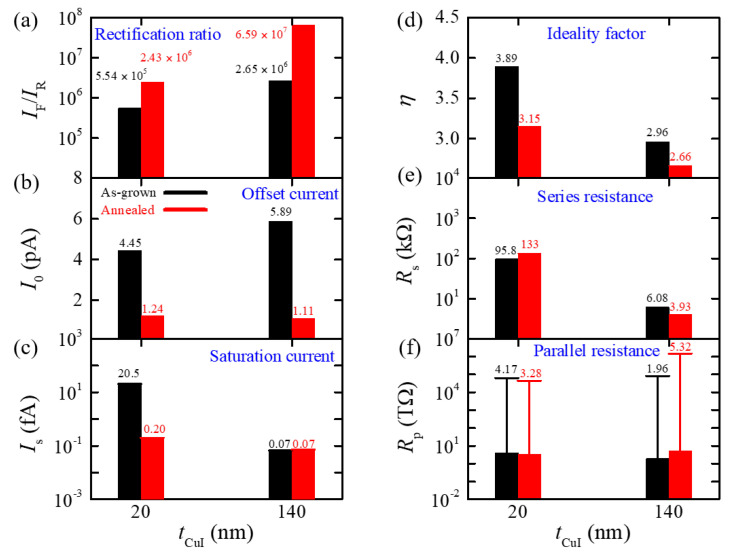
Summary of various diode parameters; (**a**) current rectification ratio *I*_F_/*I*_R_, (**b**) offset current *I*_o_, (**c**) reverse saturation current *I*_s_, (**d**) ideality factor *η*, (**e**) series resistance *R*_s_, and (**f**) parallel resistance *R*_p_. Black and red colors represent the data of the as-grown and annealed diode, respectively. All the parameters are estimated by fitting the results in Figure 2b,d with Equation (1). Error bars in (**f**) represent the least square fitting error. Error bars from the least square fitting were less than 5% for the other parameters (**b**–**e**).

**Figure 4 nanomaterials-11-01237-f004:**
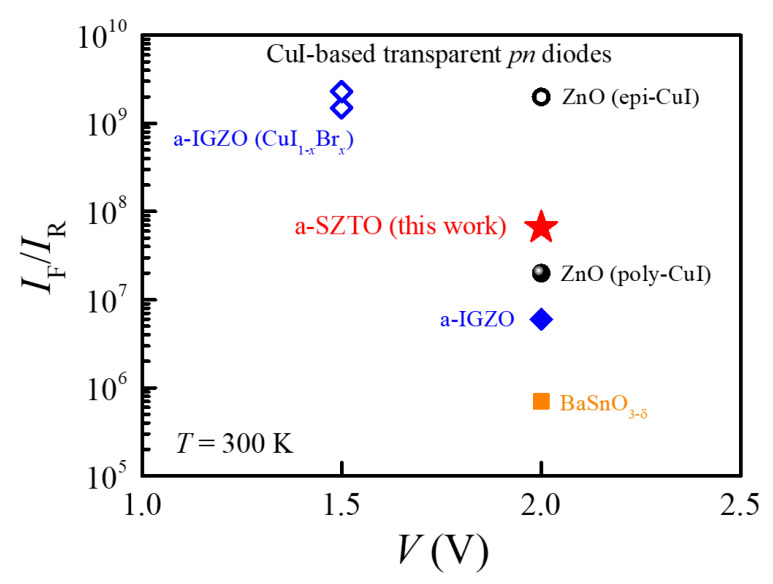
Summary of current rectification ratios *I*_F_/*I*_R_ of various heterojunction diodes consisting of the *p*-CuI film. The color indicates different *n*-type films for the *pn* diodes, and here only the cases of a high *I*_F_/*I*_R_ > ≈10^5^ are provided. *I*_F_/*I*_R_ of the CuI/SZTO diode is estimated by the fitted *I*_F_ at +2 V and *I*_R_ at −2 V in this work. *I*_F_/*I*_R_ of the CuI_1−*x*_Br*_x_*/IGZO diode is estimated by the measured *I*_F_ at +1.5 V and the fitted reverse saturation current [10]. *I*_F_/*I*_R_ of CuI/ZnO [2,11], CuI/a-IGZO [3], and CuI/BaSnO_3-δ_ [12] diodes are obtained from the measured current at ±2 V.

**Figure 5 nanomaterials-11-01237-f005:**
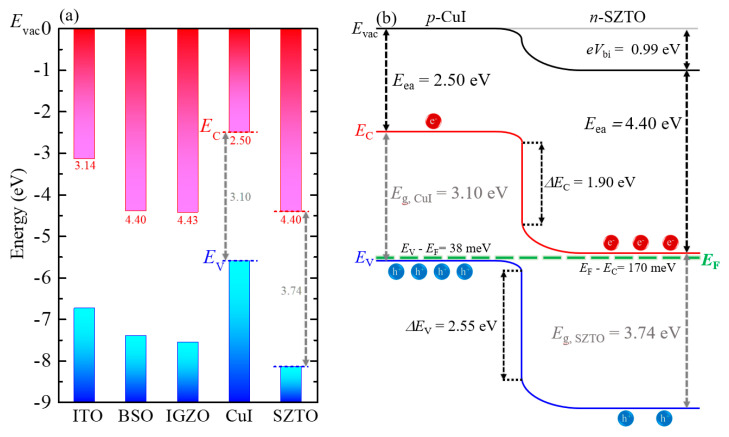
(**a**) Energy band alignment of several wide bandgap materials, including ITO, BSO, IGZO, CuI, and SZTO. The band energies of those materials are aligned at the vacuum energy level *E*_vac_ = 0. (**b**) A proposed energy band diagram of the annealed CuI (140 nm)/SZTO diode based on the Anderson heterojunction diode model, exhibiting the type-II band alignment.

**Table 1 nanomaterials-11-01237-t001:** Summary of diode parameters for various different *pn* junctions, consisting of the *p*-type CuI.

*pn* Diode	Current Rectification Ratio *I*_F_/*I*_R_	Saturation Current (A) *I*_s_	Ideality Factor *η*	Series Resistance (Ω) *R*_s_	Parallel Resistance (Ω) *R*_p_	Responsivity (Wavelength) (mA W^−1^)
pc-CuI/epi-ZnO [2]	2 × 10^7^ ± 2 V	1.3 × 10^−11^	1.7	2.9 × 10^2^	>10^12^	-
pc-CuI/epi-ZnO [2]	4 × 10^7^ ± 2 V	6.2 × 10^−15^	1.6	2.0 × 10^2^	>10^12^	-
pc-CuI/epi-ZnO [2]	2 × 10^7^ ± 2 V	3.2 × 10^−13^	1.8	2.5 × 10^2^	>10^12^	-
epi-CuI/epi-ZnO [11]	2 × 10^9^ ± 2 V	1.1 × 10^−12^	1.7	1.7 × 10^2^	3 × 10^12^	-
pc-CuI/epi-BaSnO_3−__δ_ [12]	7 × 10^5^ ± 2 V	9.9 × 10^−13^	1.5	~5.5 × 10^2^	2 × 10^9^	-
pc-CuI/a-InGaZnO [3]	6 × 10^6^ ± 2 V	-	1.6	2.0 × 10^2^	-	0.3 (365 nm)
pc-CuI_1−*x*_Br*_x_*/a-InGaZnO [10]	~2 × 10^9^ ± 1.5 V	~2 × 10^−12^	1.9	-	-	-
np-CuI/sc-CsPbBr_3_ [9]	3 × 10^2^ ± 2 V	-	-	-	-	1.4 (540 nm)
pc-CuI/sc-Ga_2_O_3_ [19]	6 × 10^3^ ± 2 V	-	3.7	-	-	2.49 (254 nm)
pc-CuI(20 nm)/a-SiZnSnO (this work)	2 × 10^6^ ± 2 V	2.0 × 10^−16^	3.2	1.3 × 10^5^	3 × 10^12^	-
pc-CuI(140 nm)/a-SiZnSnO (this work)	7 × 10^7^ ± 2 V	0.7 × 10^−16^	2.7	3.9 × 10^3^	5 × 10^12^	-

epi: epitaxial film, pc: polycrystalline film, sc: single crystal, a: amorphous, np: nanoparticle.

**Table 2 nanomaterials-11-01237-t002:** Input parameters of the Anderson heterojunction diode model for the annealed *p*-CuI/*n*-SiZnSnO (SZTO) diode where *m*_o_ is the electron mass.

Film	Bandgap (eV)	Electron Affinity (eV)	Carrier Density (cm^−3^)	Dielectric Constant	Effective Mass
*p*-CuI	3.1	^a^ 2.5	9.4 × 10^18^	^a^ 5.1	^b^ 1.4*m*_o_
*n*-SZTO	^c^ 3.7	^c^ 4.4	^c^ 6.8 × 10^15^	^d^ 4.75	^d^ 0.2*m*_o_

^a^ Ref. [2], ^b^ Ref. [28], ^c^ Ref. [13], ^d^ Ref. [22].

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
