# Peer review of "Characteristics and Electronic Band Alignment of a Transparent p-CuI/n-SiZnSnO Heterojunction Diode with a High Rectification Ratio"

_nanomaterials, 2021, doi:10.3390/nano11051237_

Round 1

Reviewer 1 Report

The authors fabricated  P-Cui/N-Siznsno heterojunction diode with a high
rectification ratio. the electrical results are interesing, however, some important evidences are missing.
The authors demonstrated that  A nitrogen annealing process reduces ionized impurity scattering dominantly incurred by Cu vacancy and structural defects at the grain boundaries in the CuI film to result in improved diode
performance. However, some direct evdiences are missiong, for example, XPS measurment or  SEM or AFM images can comfirm that the concentration of Cu vacancy  or grain boundaries after annealing.

Reviewer 2 Report

The paper is clearly written and presents a solid study onto electrical properties of the investigated p-n junctions. I have only the following minor remarks and suggested changes:

  1. sample preparation: lines 89-95: DC and RF magnetron sputtering: was it a custom made equipment or commercial? Provide more details.
  2. same for thermal evaporation: line 99.
  3. degr Celsius (for example line 95) check the way of writing it.
  4. I-V curves for Au/Ni film with CuI, line 108-11: unclear. Two tungsten probes onto two different dots each? If on the same dot, then nothing is confirmed. If between CuI and Au/Ni then semiconducting behaviour of CuI to subtract.... Equipment? A sample curve to supplementary data?
  5. Fig. 1b: a symbol "4mm" is misleading, since a dashed area is likely 4mm in width and 10 mm in lenght. Thus, size of an image is likely much more than 4mm.
  6. Fig. 1c: what does T% 85% refer to? It should rather be max. T % measured.
  7. Corresponding description of "T" measurements in text, lines 154-159: the Authors should first say that they are going to use the relationship Tfilm=Tmeas/Tglass, and then to say that from that relationship, Tfilm is about 87% for lambas 500 - 700 nm. Otherwise, it is much less.
  8. I-V curves obtaining method for Fig. 2 should be described a bit better. Two point or four point contact?
  9. What are errors of the parameters in Fig. 3. 
  10. Fig. 1 was obtained for 110 nm thick CuI film. Why Figs. 2 and 3 are for 20 nm and 140 nm thick CuI films? Why data T, R, and similar (Fig. 1) data for 140 nm or 20 nm CuI films are not presented?

Reviewer 3 Report

In this manuscript, the authors reported on the transparent p-CuI/n-SiZnSnO (SZTO) heterojunction diodes fabricated by thermal evaporation of a (111) oriented p-CuI polycrystalline film on top of an amorphous n-SZTO film grown by the RF magnetron sputtering method. In my opinion, this manuscript is interesting to the readers of Nanomaterials. The topic is important in this field. However, there are several issues needed to be improved and revised before the possible publication in Nanomaterials. A major revision is suggested.

  1. English is suggested to be polished by a native speaker to a publishable level. For heterojunction diode (a recent review by Chen et al in Adv. Funct. Mater. 2020, 30, 1909909) can be referred to. It is suggested the authors to check their manuscript carefully and thoroughly to avoid some typical mistakes and mistypes.
  2. For this topic, such as CuI thin films as UV photodetectors or FETs J. Alloys Comp. 2021, 859, 158383; J. Phys. Chem. Lett. 2019, 10, 2400-2407; Mater. Lett. 2020, 262, 127074; Adv. Mater. Interfaces 2019, 6, 1900669 etc are valuable for being referred to.
  3. More structural characterizations, such as SEM or AFM images of p-CuI polycrystalline film, amorphous n-SZTO film and p-CuI/n-SiZnSnO (SZTO) heterojunction are necessary for the readers.
  4. As there are too many reports in this field. Firstly, the introduction part is suggested to be re-organized to highlight the novelty and originality of the present work. Secondly, the readers would like to see a paragraph with a table listing all the key parameters of diodes or photodetectors near the end of the manuscript before the conclusion to dedicate to the comparison between the present work and previous reports. The information in Figure 4 is not enough for the readers. More recent literature is suggested to be included as comparison.
  5. The referee suggests enriching the discussion based on the experimental results, which will be very important for the readers in the relative field.

In general, this review seems to be interesting and the referee would like to see the revision if possible.

Round 2

Reviewer 1 Report

Most of the concerns are addressed well, and thus it is recommended to be published as it is.

Reviewer 3 Report

Most of the concerns are addressed well, and thus it is recommended to be published as it is.